# EGCG Mediated Targeting of Deregulated Signaling Pathways and Non-Coding RNAs in Different Cancers: Focus on JAK/STAT, Wnt/β-Catenin, TGF/SMAD, NOTCH, SHH/GLI, and TRAIL Mediated Signaling Pathways

**DOI:** 10.3390/cancers12040951

**Published:** 2020-04-12

**Authors:** Ammad Ahmad Farooqi, Marina Pinheiro, Andreia Granja, Fulvia Farabegoli, Salette Reis, Rukset Attar, Uteuliyev Yerzhan Sabitaliyevich, Baojun Xu, Aamir Ahmad

**Affiliations:** 1Institute of Biomedical and Genetic Engineering (IBGE), Islamabad 54000, Pakistan; ammadfarooqi@rlmclahore.com; 2LAQV, REQUIMTE, Departamento de Ciências Químicas, Faculdade de Farmácia, Universidade do Porto, 4050-313 Porto, Portugal; andreia26293@gmail.com (A.G.); shreis@ff.up.pt (S.R.); 3Department of Pharmacy and Biotechnology (FaBiT), University of Bologna, 40126 Bologna, Italy; fulvia.farabegoli@unibo.it; 4Department of Obstetrics and Gynecology, Yeditepe University, Ataşehir/İstanbul 34755, Turkey; ruksetattar@hotmail.com; 5Department of Health Policy and Health Care Development, Kazakh Medical University of Continuing Education, Almaty 050004, Kazakhstan; e.uteuliyev@ksph.kz; 6Food Science and Technology Program, Beijing Normal University-Hong Kong Baptist University United International College, Zhuhai 519087, China; baojunxu@uic.edu.hk; 7Department of Medicine, University of Alabama at Birmingham, Birmingham, AL 35205, USA; aamirahmad100@gmail.com

**Keywords:** EGCG, signaling pathways, non-coding RNAs, anti-cancer drug

## Abstract

Decades of research have enabled us to develop a better and sharper understanding of multifaceted nature of cancer. Next-generation sequencing technologies have leveraged our existing knowledge related to intra- and inter-tumor heterogeneity to the next level. Functional genomics have opened new horizons to explore deregulated signaling pathways in different cancers. Therapeutic targeting of deregulated oncogenic signaling cascades by products obtained from natural sources has shown promising results. Epigallocatechin-3-gallate (EGCG) has emerged as a distinguished chemopreventive product because of its ability to regulate a myriad of oncogenic signaling pathways. Based on its scientifically approved anticancer activity and encouraging results obtained from preclinical trials, it is also being tested in various phases of clinical trials. A series of clinical trials associated with green tea extracts and EGCG are providing clues about significant potential of EGCG to mechanistically modulate wide ranging signal transduction cascades. In this review, we comprehensively analyzed regulation of JAK/STAT, Wnt/β-catenin, TGF/SMAD, SHH/GLI, NOTCH pathways by EGCG. We also discussed most recent evidence related to the ability of EGCG to modulate non-coding RNAs in different cancers. Methylation of the genome is also a widely studied mechanism and EGCG has been shown to modulate DNA methyltransferases (DNMTs) and protein enhancer of zeste-2 (EZH2) in multiple cancers. Moreover, the use of nanoformulations to increase the bioavailability and thus efficacy of EGCG will be also addressed. Better understanding of the pleiotropic abilities of EGCG to modulate intracellular pathways along with the development of effective EGCG delivery vehicles will be helpful in getting a step closer to individualized medicines.

## 1. Introduction

Genomic approaches such as whole genome sequencing and genetic mapping have helped considerably in the identification of many genetic variants in multiple components of cell signaling pathways. Moreover, advancements in functional genomics have marked a new frontier in molecular oncology. Epigallocatechin-3-gallate (EGCG) is a phenolic compound present in tea and has captivated tremendous attention in the past two decades because of its premium pharmacological properties. There is a wide variety of reviews published with reference to EGCG mediated anticancer effects [1,2,3,4]. However, in this review we focused on EGCG mediated modulation of deregulation cell signaling pathways in different cancers. We partitioned this multi-component review into different sections. We will open the review by critical analysis of layered regulation of the JAK-STAT pathway by EGCG. 

## 2. Targeting of JAK/STAT Signaling

The JAK-STAT pathway constitutes a rapid membrane-to-nucleus signaling module that has been shown to play fundamental role in cancer development and progression (shown in Figure 1). In this section, we will discuss in detail how EGCG modulated JAK/STAT signaling. EGCG has been shown to interfere with the JAK/STAT pathway at different steps, which includes inhibition of STAT phosphorylation and restriction of nuclear transportation of STAT proteins. 

EGCG remarkably reduced tyrosine and serine phosphorylation of signal transducer and activator of transcription 1 (STAT1) [5]. Moreover, phosphorylation of protein kinase C delta PKC-delta, Janus kinase 1 (JAK1), and Janus kinase 2 (JAK2), which are the upstream activators of STAT1 are also inhibited by EGCG in interferon gamma (IFNγ)-stimulated oral cancer cells (shown in Figure 1) [5]. EGCG-mono-palmitate (EGCG-MP), a highly active derivative of EGCG effectively activated Src homology 2 domain-containing tyrosine phosphatase-1 (SHP-1) which consequentially resulted in reduction of phosphorylated levels of BCR-ABL and signal transducer and activator of transcription 3 (STAT3) in human chronic myeloid leukemia (CML) cells (shown in Figure 1) [6]. EGCG-MP treatment more efficiently induced regression of tumor growth in BALB/c athymic nude mice [6]. EGCG potently inhibited BCR/ABL oncoprotein and the JAK2/STAT3/AKT pathway in BCR/ABL+ CML cell lines [7]. Curcumin worked synchronously with EGCG and considerably interfered with tumor conditioned media-induced transition of normal endothelial cells toward tumor endothelial cells by inhibition of the JAK/STAT3 signaling pathway [8].

EGCG significantly reduced phosphorylation of STAT3 on the 705th tyrosine residue and improved sensitivity of cisplatin-resistant oral cancer cells [9]. Fundamental role-play of STAT signaling had previously been studied in invasive breast cancers and matched lymph nodes using quantitative immunofluorescence [10]. STAT proteins were analyzed in lymph nodes and paired primary breast cancer tissues. There was higher expression of cytoplasmic STAT1, p-STAT3 (Ser727), STAT5, and nuclear p-STAT3 (Ser727) in the nodes [10]. c-Myb overexpression induced activation of NF-κB and STAT3 signaling to enhance proliferation, invasion, and resistance against cisplatin [11]. However, c-Myb silencing inhibited proliferation, invasive potential, and sensitized ovarian cancer cells to cisplatin. EGCG completely inhibited c-Myb-mediated proliferative and invasive abilities of ovarian cancer cells [11].

EGCG dose-dependently reduced phosphorylated levels of STAT1 and STAT3 [12]. Quercetin and EGCG worked synergistically and exerted inhibitory effects on cytokine-mediated upregulation of iNOS (inducible nitric oxide synthase) and ICAM-1 (intercellular adhesion molecule-1) via JAK/STAT cascade in cholangiocarcinoma cells (Figure 1) [12].

Indoleamine 2,3-dioxygenase (IDO) is a tryptophan catabolic enzyme. IDO mechanistically regulates immunological response and enables tumor cells to evade the immune system [13]. IFN-γ increased mRNA and protein levels of IDO in HT29 and SW837 colorectal cancer cells. EGCG dose-dependently decreased IFN-γ-induced expression of IDO in SW837 cells. Increase in p-STAT1 level induced by IFN-γ was also found to be markedly repressed by EGCG. Data obtained from reporter assays clearly revealed that EGCG inhibited the transcriptional activity of IDO promoter and blocked binding of p-STAT1 to gamma-activated sequence (GAS) sites on the promoters of target genes (Figure 1) [13]. 

Toxicological analysis of EGCG highlighted its efficacy and minimum off-target effects. Orally administered EGCG mitigated cisplatin-induced hearing loss along with a marked reduction in the loss of outer hair cells in the basal cochlear region. Importantly, chemotherapeutic drug-induced toxicity was also reduced mainly though suppression of apoptotic markers and oxidative stress [14].

It has recently been reported that IFNγ-mediated PD-L1 levels were noted to be downregulated after treatment with green tea extracts and EGCG mainly through inhibition of JAK2/STAT1 signaling in A549 cells [15]. Likewise, EGF-stimulated PD-L1 upregulation was reduced in EGCG-treated Lu99 cells by inactivation of EGFR/AKT transduction cascade. Additionally, green tea extracts notably reduced average number of tumors and percentage of PD-L1^+^ cells in lungs of A/J mice intraperitoneally injected with a cigarette smoke toxin. EGCG reduced mRNA levels of PD-L1 in F10-OVA cells and enhanced expression of interleukin-2 in tumor-specific CD3^+^ T cells [15]. Collectively these findings suggested that green tea catechin acted as a useful immunological checkpoint inhibitor.

Confluence of information suggested central role of JAK/STAT signaling in different cancers. EGCG mediated inhibition of JAK/STAT signaling via activation of negative regulators (SHP-2) and inactivation of positive regulators (JAK1, JAK2) has gradually gained appreciation. Additionally, different fusion oncoproteins (BCR-ABL) are also exclusively targeted by EGCG. 

## 3. VEGF/VEGFR Signaling

EGCG and silibinin worked synergistically and inhibited vascular endothelial growth factor/vascular endothelial growth factor receptor (VEGF/VEGFR) signaling. EGCG and Silibinin also reduced migratory potential of A549 cells [16]. EGCG interacted with VEGF mainly through hydrophobic interactions and induced a change in the secondary structure of the protein (Figure 1) [17].

Vandetanib (ZD6474), a VEGFR inhibitor was co-loaded with EGCG in mesoporous Silica-Gold nanoclusters for effective targeting of tamoxifen-resistant breast cancer cells [18]. Vandetanib and EGCG effectively reduced phosphorylated levels of EGFR2 and VEGFR2 in drug-resistant breast cancer cells [18]. EGCG also worked with superior efficacy when used in combination with tamoxifen. Tamoxifen worked powerfully with EGCG and reduced the levels of EGFR1, VEGF, and VEGFR1 in breast cancer cells [19]. SU5416 (Semaxanib) also worked remarkably with EGCG and induced apoptosis in malignant neuroblastoma SK-N-BE2 and SH-SY5Y cells [20]. SU5416 and EGCG also inhibited VEGFR2 expression [20].

EGCG dose-dependently decreased levels of VEGFR2 and p-VEGFR2 in HCC and colorectal cancer cells (Figure 1) [21,22]. EGCG induced regression of tumors in mice xenografted with either HuH7 or SW837 cells. EGCG decreased total and phosphorylated levels of VEGFR2 in these xnografts [21,22].

Detailed mechanistic insights revealed that p-STAT1 and p-STAT3 formed complexes with VEGFR1 and VEGFR2 in chronic lymphocytic leukemia (CLL) cells [23]. VEGF induced nuclear accumulation of p-STAT3 in primary CLL B cells. VEGF/VEGFR complex facilitated shuttling of STAT3 from the plasma membrane to perinuclear regions. VEGF induced co-localization of STAT3, VEGFR1 and VEGFR2 to the same perinuclear regions. Collectively these findings provided clear evidence that the VEGF/VEGFR pathway “switched on” STAT proteins which induced resistance against apoptosis. EGCG decreased levels of p-STAT3 [23]. EGCG also remarkably reduced phosphorylated levels of VEGFR1 and VEGFR2 in B-cell chronic lymphocytic leukemia cells [24].

## 4. Regulation of Methylation-Associated Machinery 

PRC2 (Polycomb repressive complex-2) is a transcriptional repressive complex that consists of three essential proteins: EZH2 (enhancer of zeste-2), EED (embryonic ectoderm development), and SUZ12 (suppressor of zeste 12). A series of structural studies have shown that EZH2 context-dependent trimethylates lysine 27 on histone 3 (H3K27) to promote transcriptional inactivation of target genes (shown in Figure 2). 

EZH2-mediated trimethylation of H3K27 induced transcriptional repression of TIMP3 (tissue inhibitor of metalloproteinases-3). However, EGCG demonstrated remarkable ability to inhibit EZH2-mediated trimethylation. There was a considerable reduction in the levels of enhancers of zeste homolog 2 (EZH2) and H3K27me3 repressive marks at the promoter region of TIMP-3. Additionally, there was an evident increase in histone H3K9/18 acetylation [25]. Essentially, green tea polyphenols and EGCG treatment significantly reduced class I histone deacetylases (HDAC) activity/expression in prostate cancer cells. Furthermore, levels of EZH2 and H3K27me3 were also found to be reduced in prostate cancer cells [25]. Data clearly suggested that EGCG efficiently demonstrated multi-layered regulation of HDACs and EZH2. 

Due to the fundamental role of EZH2 in cancer progression, different inhibitors of EZH2 have been designed and tested for evaluation of efficacy. EGCG and GSK343 (EZH2 inhibitor) exerted inhibitory effects on the proliferation, invasive and migratory potential of the cells, and suppressed EZH2-mediated trimethylation of H3K27 [26].

Recent advancements in the biochemical characterization of polycomb-group (PcG) complexes have revealed a broad range of new proteins, which assemble to form multi-protein complexes. All PRC1 complexes contain Ring1B, which has the E3 ubiquitin ligase activity of the complex. Complexes also include PCGF4/BMI-1 in association with Ring1B to regulate epigenetic modifications [27]. EGCG reduced BMI-1 and EZH2 levels in SCC-13 cells [28].

PML–RARα homodimers worked synchronously with co-repressors and histone deacetylases (HDACs) and consequentially enhanced DNA methylation [29]. EGCG reduced the levels of HDAC1 and PML/RARα in leukemic cells (Figure 2) [30].

Groundbreaking discoveries in biology of epigenome have enabled us to develop a sharp comprehension of highly intricate and well-coordinated interplay of HDACs, histone methyltransferases, and DNA methyltransferases. EGCG has emerged as a master-regulator of epigenetic-associated machinery. 

Chromatin immunoprecipitation (ChIP) analyses revealed that EGCG enhanced hyperacetylated H4 and acetylated H3K14 histones within promoter regions of p27, PCAF, C/EBP and reduced binding of PRC2 core component genes EZH2, SUZ12, and EED [31]. 

EGCG significantly reduced enzymatic activities of DNA methyltransferase (DNMT) and HDAC in HeLa cells [32]. Moreover, EGCG also reduced expression level of DNMT3B whereas expression levels of HDAC1 remained unchanged [32]. GTP/EGCG-promoted acetylation of p53 and enhanced its binding to the promoters of Bax and p21/waf1. Treatment of cells with GTPs and EGCG dose- and time-dependently inhibited class I HDACs [33].

Am80 is a structurally different synthetic retinoid from all-trans-retinoic acid. EGCG and Am80 increased acetylated-p53 and acetylated-α-tubulin through suppression of HDAC activity. Use of specific inhibitors against HDAC4 and HDAC5 strongly induced p21waf1 gene expression. Additionally, HDAC6 inhibition induced upregulation of GADD153 and p21waf1 [34].

UHRF1 (ubiquitin-like containing PHD and Ring finger 1) contributed to inactivation of tumor suppressor genes by directing the binding of DNA methyltransferase 1 (DNMT1) to hemi-methylated promoters [35]. EGCG downregulated DNMT1 and UHRF1 expression and consequently upregulated p73 and p16 (INK4A) in Jurkat cells. UHRF1 downregulation was dependent upon the generation of ROS by EGCG. Upregulation of p16 (INK4A) correlated strongly with reduction in the binding of UHRF1 to the promoter region. UHRF1 overexpression counteracted EGCG-induced apoptosis and upregulation of p73 and p16 (INK4A) [35].

EGCG effectively reduced 5-methylcytosine, DNMT activity, mRNA and protein levels of DNMT1, DNMT3a, and DNMT3b [36]. EGCG decreased HDAC activity and increased levels of acetylated H3K9 and H3K14, H4K5, H4K12, and H4K16 but decreased levels of methylated H3-Lys 9. Collectively, because of inhibition of DNMTs and HDACs, EGCG induced re-expression of p16INK4a and Cip1/p21 [36].

Gazing through a molecular lens clearly highlighted contextual push and pull between various versatile regulators associated with methylation. Substantial fraction of information gathered through high-quality research has unraveled that a broad range of tumor suppressors are epigenetically silenced during cancer progression. Selective targeting of DNMTs and HDACs specifically in cancer cells is very challenging and needs to be comprehensively investigated in EGCG-treated preclinical models. In the upcoming section we will analyze how EGCG modulated deregulated TGF/SMAD signaling. 

## 5. TGF/SMAD Signaling 

Binding of TGFβ superfamily ligands to a type II receptor facilitated closer positioning of type I receptor and phosphorylated it [37]. More importantly, type-I receptor mediated phosphorylation of receptor-regulated SMADs (R-SMADs), which promoted formation of a complex with common mediator SMAD (co-SMAD) (shown in Figure 3). Structural studies had shown that the R-SMAD/co-SMAD complex accumulated in the nucleus to transcriptionally modulate the expression of target genes [38]. Epithelial to mesenchymal transition (EMT) is a highly complex mechanism induced by TGF/SMAD signaling. SMAD2/3 proteins have been shown to stimulate the expression of Snail and Slug in different cancers [39]. 

In this section, we will provide an overview of multi-layered regulation of TGF/SMAD signaling by EGCG in different cancers. Inhibition of phosphorylation of R-SMADs will inhibit TGF/SMAD signaling. Consequentially, TGF/SMAD signaling inhibition will result in repression of EMT-associated markers. 

EGCG effectively reduced p-SMAD3, Snail, and Slug levels in ovarian cancer cells [40]. EGCG considerably suppressed EMT, invasive and migratory capacity of anaplastic thyroid carcinoma (ATC) 8505C cells by regulation of the TGFβ/SMAD pathway [41]. EGCG exerted inhibitory effects on TGFβ1-induced expression of EMT markers (vimentin) in 8505C cells. EGCG was noted to completely block the phosphorylation of SMAD2/3 and nuclear accumulation of SMAD4 [41].

Apart from phosphorylation, acetylation of SMAD proteins is also an intricate mechanism. p300/CBP, a histone acetyltransferase, has been shown to post-translationally modify SMAD proteins. TGFβ1-driven activation of p300/CBP mediated EMT mainly through acetylation of SMAD2 and SMAD3 [42]. EGCG inhibited p300/CBP activity in lung cancer cells. EGCG strongly repressed TGFβ1-induced EMT and reversed the upregulation of different target genes associated with EMT. EGCG inhibited TGFβ1-mediated activation of p300/CBP. EGCG inhibited TGFβ1-mediated EMT by interfering with the acetylated state of SMAD2 and SMAD3 in lung cancer cells [42].

TGFβ potently induced epithelial–mesenchymal transition (EMT) in NSCLC cells but EGCG reversed TGFβ-induced morphological alterations [43]. EGCG upregulated the expression of E-cadherin and downregulated the expression of vimentin. Data obtained through immunofluorescence also provided clear clues that E-cadherin was upregulated, and vimentin was downregulated by EGCG [43]. Moreover, EGCG effectively inhibited TGFβ-induced migratory and invasive potential of NSCLC cells. EGCG inhibited TGFβ-induced EMT at the transcriptional level. Expectedly, EGCG reduced phosphorylated levels of ERK1/2 (extracellular signal-regulated protein kinases 1/2) and SMAD2 and also inhibited the accumulation of SMAD2 in the nucleus. EGCG repressed the expression of transcriptional factors Slug, Snail, Twist, and ZEB1 and upregulated E-cadherin expression (Figure 3) [43]. 

Interestingly, different peptide aptamers have been designed to effectively inhibit interaction of SMAD2 and SMAD3 with SMAD4. Therefore, it might be advantageous to combine EGCG with different TGFβ signaling inhibitors to inhibit tumor growth in xenografted mice. More importantly, it will also be exciting to evaluate EGCG-mediated regulation of negative regulators (SMURFs and NEDDs) of the TGF/SMAD pathway. 

## 6. Regulation of Wnt/β-Catenin Pathway

Detailed mechanistic insights revealed that in the absence of Wnt signal, β-catenin was phosphorylated by APC (adenomatous polyposis coli)/Axin/GSK3β complex and degraded by proteasome [44]. However, activation of the membrane receptors by Wnt signal resulted in the phosphorylation and degradation of GSK3β. EGCG inhibited phosphorylation of GSK3β, upregulated GSK3β expression, and decreased the levels of β-catenin in colorectal cancer cells [44].

O^6^-methylguanine (O6-meG) DNA-methyltransferase (MGMT) is a versatile mediator of temozolomide resistance in glioblastomas. TCF/LEF-binding sites within the promoter region of the MGMT gene have previously been identified [45]. Intriguingly, there is evidence of regulation of MGMT by WNT/β-catenin signaling. EGCG not only prevented translocation of β-catenin into the nucleus but also reduced the levels of transcriptional factors TCF1 and LEF1 [46]. Overall these findings clearly suggested that EGCG repressed MGMT expression via inhibition of the β-catenin-driven pathway. 

EGCG not only reduced mRNA levels and transcriptional activities of β-catenin in p53 wild-type-expressing KB cells but also promoted ubiquitylation and degradation of β-catenin [47]. EGCG dose-dependently suppressed β-catenin expression in MDA-MB-231 cells [48]. EGCG worked synergistically with gemcitabine and exerted stronger inhibitory effects on β-catenin and N-cadherin in pancreatic cancer cells [49]. 

Clinical trials of CWP232291 (NCT01398462) and PRI-724 (NCT01302405; NCT01764477) are providing important clinically relevant information and it will be interesting to combine these agents with EGCG for evaluation of robust inhibition of β-catenin-driven signaling and tumor growth inhibitory effects in xenografted mice. 

## 7. Regulation of Notch Pathway 

The Notch signaling pathway consists of the Notch receptor, Notch ligand, DNA-binding protein, and Notch regulatory molecules. Notch is a transmembrane protein that mediates communication between neighboring cells. Binding of the ligands to a Notch receptor promoted proteolytic cleavage of NICD (Notch intracellular domain) and its consequential nuclear translocation where it complexed with CSL and formed NICD/CSL transcriptional activation assembly for stimulation of HES and HEY.

EGCG decreased mRNA levels of Notch1, Hey1, and Hes1 [50]. Western blot assay clearly indicated that EGCG dose-dependently reduced protein levels of Notch1 in cancer stem cells (CSCs) of head and neck squamous carcinoma (HNSC). Additionally, EGCG dose-dependently decreased the Notch promoter activity [50].

Tumor growth was significantly reduced in HuCC-T1 tumor-bearing mice subcutaneously injected with EGCG. Notch1 was found to be markedly reduced by EGCG treatment [51]. 

Expression levels of Hes1 and Notch2 were observed to be considerably reduced in EGCG treated colorectal cancer cells [52]. More importantly, EGCG inhibited Notch1 and cleaved-Notch1 in 5-fluorouracil-resistant colorectal cancer cells [53]. 

## 8. Regulation of TRAIL Mediated Apoptosis 

Increasingly it is being realized that cancer cells harbor highly complex signaling networks that resist apoptotic programming. High-quality research works related to the TRAIL-mediated pathway in the past two decades have ignited the field of molecular oncology and yielded a stream of preclinical and clinical insights that have reshaped current knowledge of apoptotic cell death.

GCG and TRAIL synergistically reduced Bcl-XL, Bcl-2, and FLIP. Whereas, combinatorial treatment activated capase-8, -9, and -3 in nasopharyngeal carcinoma cells [54]. 

EGCG and TRAIL also worked effectively against renal cell carcinoma and pancreatic cancer cells [55,56].

EGCG restored sensitivity of HCC cells to TRAIL-induced apoptosis [57]. EGCG upregulated caspase-3 activity and simultaneously downregulated Bcl-2 levels. EGCG also induced upregulation of DR4 and DR5 [57]. EGCG and TRAIL robustly enhanced DR4 levels. Furthermore, FLIP levels were reduced in prostate cancer cells treated in combination with EGCG and TRAIL [58]. Collectively these findings suggested that EGCG might be helpful in increasing the protein levels as well as cell surface expression of death receptors. There is sufficient experimental evidence related to reduction in the cell surface levels of death receptors. Death receptors are internalized and degraded in various cancers. Therefore, EGCG might play its role in stabilizing the levels of death receptors. 

PEA15 (phosphoprotein-enriched in astrocytes) is an oncoprotein [59]. It has previously been reported that AKT-induced PEA15 phosphorylation and increased its stability. EGCG downregulated PEA levels mainly through inactivation of AKT. However, overexpression of PEA15 severely impaired apoptotic cell death induced by EGCG and TRAIL [59].

Certain hints had emerged which highlighted that EGCG inhibited TRAIL-induced apoptosis and activated autophagic flux in colon cancer cells. Inhibition of autophagic flux induced death receptor-driven apoptosis in colon cancer cells [60].

These scientific findings are intriguing and future research must converge on identification of additional protein targets in the TRAIL-driven pathway. Essentially, the TRAIL mediated pathway is regulated by long non-coding RNAs as well. Therefore, it will be paramount to unravel underlying mechanisms of TRAIL resistance and identification of proteins, which can be targeted to restore apoptosis in TRAIL-resistant cancers. Keeping in view the fact that TRAIL-based therapeutics and death receptor-targeting agonistic antibodies have entered into various phases of clinical trials, any progress in improving the efficacy of TRAIL-based therapeutics will be advantageous. 

## 9. Regulation of Non-Coding RNAs by EGCG in Different Cancers

Discovery of non-coding RNAs has revolutionized our current understanding of the mechanisms that regulate post-transcriptional processes. The field of non-coding RNA has been extensively explored and researchers have witnessed groundbreaking advancements in disentangling the complicated web ranging from biogenesis of non-coding RNAs to post-transcriptional regulation of a myriad of target mRNAs. 

A wealth of information has unveiled a fundamental role of non-coding RNAs in different cancers and researchers have experimentally verified the effects of natural and synthetic products on wide-ranging microRNAs and long non-coding RNAs in different cancers. 

### 9.1. Tumor Suppressor miRNAs

miR-485, a tumor suppressor microRNA, has been found to be frequently downregulated in various cancers. CD44 was directly targeted by miR-485 in A549-cisplatin resistant lung cancer cells. CD44 was overexpressed in A549-cisplatin resistant lung cancer cells but EGCG treatment exerted repressive effects on CD44 levels by enhancing miR-485-mediated targeting of CD44 [61]. EGCG also induced regression of tumors in mice xenografted with A549-cisplatin resistant lung cancer cells. 

miR-485-5p directly targeted RXRα in drug-resistant lung cancer cells. EGCG repressed CSC-like properties via modulation of the miR-485-5p/RXRα axis [62]. miR-155 is a tumor suppressor miRNA reportedly involved in enhancing drug sensitivity of cancer cells [63]. EGCG promoted NF-κB mediated upregulation of miR-155. Resultantly, miR-155 enhanced drug sensitivity of colorectal cancer cells by directly targeting MDR1 [63]. IGF2BP1 and IGF2BP3 are direct targets of miR-1275 in different cancers [64]. EGCG stimulated the expression of miR-1275 and potentiated targeting of IGF2BP1 and IGF2BP3 by miR-1275 in HCC cells [65].

### 9.2. Oncogenic miRNAs 

miR-221/222 played a central role in drug resistance. Knockdown of miR-221/222 inhibited proliferation of drug-resistant breast cancer cells [66]

Suberoylanilide hydroxamic acid (SAHA), an HDAC inhibitor worked effectively with EGCG and markedly reduced expression of miR-221/222 in triple-negative breast cancer cells [67].

### 9.3. Targeting of Oncogenic LncRNAs

SOX2OT variant 7 effectively promoted Notch3/DLL3 signaling in osteosarcoma stem cells (OSCs) [68]. NOTCH target genes HEY1 and HES1 were found to be notably enhanced in variant 7-expressing OSCs. EGCG efficiently inhibited the levels of HEY1 and HES1 in OSCs. However, EGCG mediated inhibitory effects were noted to be impaired in variant 7-expressing cells [68]. EGCG mediated tumor regression was not observed in mice xenografted with variant 7-expressing OSCs. However, EGCG treatment and NOTCH3 knockdown induced reduction in tumor growth in mice inoculated with variant 7-expressing OSCs [68]. 

EGCG also downregulated lncRNA-AF085935 in HCC cells. It was suggested that lncRNA-AF085935 promoted proliferation of HCC cells [69]. However, the study did not clearly provide a link between lncRNA-AF085935 and its targets and how lncRNA-AF085935 regulated apoptosis and proliferation in HCC cells. 

### 9.4. Tumor Suppressor LncRNAs

EGCG had been shown to induce the expression of cisplatin transporter CTR1 (copper transporter 1) in cancer cells [70]. EGCG upregulated CTR1 and enhanced accumulation of intracellular platinum in NSCLC cells. hsa-miR-98-5p suppressed CTR1, whereas NEAT1 (nuclear enriched abundant transcript 1) enhanced its expression. hsa-miR-98-5p had specific complementary binding sites for NEAT1. Essentially, NEAT1 acted as a competitive endogenous RNA and upregulated EGCG-triggered CTR1 by sponging away hsa-miR-98-5p in NSCLC cells [70].

It seems surprising to note that available scientific proof related to regulation of non-coding RNAs by EGCG is limited. Keeping in view the wealth of information about remarkable pharmacological properties of EGCG, it is paramount to uncover how EGCG modulated different miRNAs, lncRNA, and circular RNAs in different cancers. Identification of the list of tumor suppressor and oncogenic non-coding RNAs regulated by EGCG in different cancers will be highly valuable in combinatorial treatments. 

## 10. Nanotechnological Approaches for Effective Delivery of EGCG to the Target Sites

Despite the ability of EGCG to modulate several cancer-related mechanisms there are still major hurdles for the establishment of EGCG in clinical settings. The therapeutic concentrations of EGCG (between 1 and 10 µM) in the majority of the studies are much higher than the concentrations monitored in the plasma or tissues of animals or in human plasma (usually lower than 1 µM) after tea ingestion. In fact, even after the consumption of 7–9 cups of tea the EGCG concentration in plasma was still lower than 1 µM [71] and for that reason the use of nanotechnology, particularly the development of nanoparticles (NPs) as drug delivery systems, represent a promising approach to increase the bioavailability of EGCG. Nanotechnology corresponds to the science that studies and creates materials with dimensions between 1 and 1000 nm. NPs have at least one of the dimensions in the nanoscale range [72]. There are several types of NPs and for more comprehensive and detailed information the reader can consult the following revisions [73,74,75,76]. The different properties of the NPs can be used for medical purposes. Due to their small scale, NPs are excellent drug carriers, and since they can be modified in various factors such as size, chemical composition, outer layer, and others they are very versatile [77]. Furthermore, NPs can modify the pharmacokinetics and the stability of the carrier compound, being, for that reason, a promising strategy to improve EGCG bioavailability profile [78]. Another interesting characteristic of NPs is the possibility to enhance the cellular uptake or even the cellular targeting by modifying the outer layer with different ligands expressed in the target cells to assign particular characteristics in a strategy known by active targeting [79]. This is a useful strategy to improve the bioavailability and stability of EGCG even further, enhancing the utilization options and ultimately enhancing the anti-cancer properties of EGCG. The main types of NPs used for the delivery of EGCG reported in the literature are gold, polymeric, lipid-based, and inorganic NPs (shown in Figure 4). The majority of the NPs are designed to be at the range of approximately 200 nm since this size allows the administration of the NPs by the oral and intravenous routes. Other types of NPs were also used for the encapsulation of EGCG for the purpose of cancer therapy, including carbohydrates, transition metals, and inorganic materials [80,81,82]. The use of targeting ligands further increased cancer cell specificity and improved the anti-tumor effects of EGCG and, for that reason, folic acid has been used frequently to functionalize the NPs, since the folic acid receptor is overexpressed in tumor cells. However, other ligands can also be used, including antibodies, carbohydrates, or polysaccharides and other molecules [83]. A summary of the studies using different EGCG nanocarriers for cancer management and carried out in cell lines and in animals is depicted in Table 1.

Gold NPs as EGCG delivery systems have been exploited in several types of cancer since gold has anti-cancer properties per se [85,100]. Several reports have described the in vitro and in vivo efficacy of gold NPs in conjugation with EGCG for cancer treatment, including for the bladder, melanoma, neuroblastoma, and hepatocarcinoma [84,85,86,87]. These nanocarriers also demonstrated a high biocompatibility, inducing low damage to human red blood cells and therefore no toxicity for the dose tested was observed. The NPs made of polymers approved and recognized as safe by the US Food and Drug Administration (FDA) are also suitable for cancer applications [89,101]. Several groups have already encapsulated EGCG into different polymeric NPs for cancer therapy, including for the treatment of prostate cancer, colorectal, breast cancer, melanoma, and gastrointestinal cancer [88,89,90,91,92,93,94,95]. Despite the high toxicity towards cancer cells these NPs demonstrated absence of toxicity for normal cells. Liposomes and lipid NPs are lipid-based NPs in composition and for that reason are biodegradable and present minimal levels of toxicity [97]. There are some studies reporting the use of lipid-nanocarriers for the delivery of EGCG to cancer cells [96,97,98,99]. All of the studies were used for the treatment of breast cancer with results that demonstrated efficacy and security in vitro and in vivo, including in the MDA-MB-231 cell line, which is a model of the triple-negative cancer and considered more aggressive and associated with poorer outcome than other types of breast cancer. 

## 11. Potential Clinical Applications

EGCG drug delivery systems based in NPs might represent an extraordinary resource to improve the application of EGCG in chemoprevention or to introduce the use of EGCG in the therapy of cancer. The idea of using drug delivery systems, such as NPs for loading EGCG, preserving its structure, and allowing to circumvent the limitations of the low bioavailability associated with the oral administration of free EGCG has a tremendous potentiality since increasing the amount of EGCG inside the cells will potentialize the effect of EGCG in the molecular targets and the effect of deregulated oncogenic signaling cascades and, therefore, determine better cancer outcomes in comparison with free EGCG. For instance, EGCG loaded in polylactic acid–polyethylene glycol NPs preserved the biological activity and efficacy on molecular targets in vitro and in xenograft tumors with over 10-fold dose advantage in comparison with EGCG alone [91]. Indeed, in vitro and in vivo studies are mandatory to verify whether EGCG loaded in NPs maintain EGCG mechanism of action and to understand if the efficacy on molecular targets is at least retained or increased. In view of a safe application, the toxicity of engineered NPs associated with EGCG needs to be fully investigated. For instance, transition metal oxide NPs have been found to increase oxidative stress, disturb calcium homeostasis, and deregulate cell cycle [102]. The activation of the immune system, specifically macrophage activation and cytokine release has been also reported [103]. Thus, lipid-based NPs show higher level of biocompatibility and bioavailability, emerging as the best candidates for pharmaceutical and clinical applications. In this context, EGCG loaded solid lipid NPs as an oral delivery system did not show any toxicity in rats [104]. Different nanoformulations, including EGCG, also showed great biocompatibility with no or very modest toxicity in animal models [105,106]. All these findings encourage the efforts to invest in biocompatible EGCG NPs to be used on humans, as interventional studies in pre-cancerous lesions, including prostate, breast, colon, and Barret’s Esophagus [107,108,109,110] demonstrated EGCG efficacy despite the poor bioavailability and low plasma concentrations. Therefore, EGCG NPs are expected to improve the chemopreventive effects and to widen the applications in pre-neoplastic lesions, where the results were unclear or incomplete. In addition, EGCG mechanism of action can be improved by the association with anti-cancer drugs already used in cancer treatment since numerous drugs used in cancer therapy, including doxorubicin, 5-flurouracile, cisplatin, paclitaxel, act synergistically with EGCG [111], the best combinations being predictable on the basis of in vitro and in vivo studies. Lastly, active targeting also represents a strategy to preferentially address NPs to cancer cells. Nanomedicine-based therapy is at the beginning, but in the context of cancer chemoprevention and therapy, EGCG NPs might become a powerful strategy over the conventional chemotherapy approach.

### Clinical Trials Evaluating EGCG

Given the promising reports from preclinical studies, EGCG has been tested in various clinical studies. Postmenopausal women are at high risk of developing breast cancer and, therefore, EGCG safety clinical trials have been conducted targeting this population. EGCG can afford benefit in terms of regulating LDL-cholesterol as well as glucose and insulin, as reported by a double-blind, randomized, placebo-controlled intervention study in healthy postmenopausal women [112]. A subsequent ancillary study of a double-blind, randomized, placebo-controlled, parallel-arm trial further confirmed the benefit of EGCG but reported the total cholesterol levels reductions only in women with elevated baseline levels [113]. In postmenopausal women, a daily dose of 843 mg EGCG has been reported to be generally well-tolerated with only a small fraction (6.7%) of women reporting adverse events [114]. This dose of 843 mg EGCG, when administered for a year, can reduce mammographic density in relatively younger women (50–55 years) but not in postmenopausal women, as suggested by phase II trial [115]. Not only in breast cancer patients or women at high breast cancer risk, EGCG is well-tolerated by chronic lymphocytic leukemia (CLL) patients as well [116]. Further, EGCG, at a daily dose as low as 44.9 mg for 4 weeks prior to surgery, has been reported to result in increased bioavailability, including accumulations in breast tumor tissue, in early stage breast cancer patients [108].

A randomized trial reported no reduction in likelihood of prostate cancer in men with high-grade prostatic intraepithelial neoplasia, compared to placebo, after a year on 400 mg EGCG dose per day [117]. It is possible that this might be related to the dose tested in this study as a previous study which tested the effects of 800 mg EGCG administered to 26 patients with positive prostate biopsies reported significant reductions in PSA, HGF, and VEGF, with no associated liver toxicity [118]. Similarly, a phase II pharmacodynamic prevention trial in bladder cancer patients indicated a possible reduction in PCNA and clusterin levels upon 2–4 weeks administration of EGCG prior to transurethral resection of bladder tumor or cystectomy [119].

EGCG has been tested in cancer clinical trials not just for the direct anticancer effects, but also for possible effects on co-morbidities. In lung cancer patients with an unresectable stage III disease, a phase I study was conducted to evaluate the efficacy of EGCG against chemotherapy related esophagitis [120]. Patients, divided in six cohorts receiving six different doses of EGCG, were administered EGCG once grade 2 esophagitis occurred. The study reported dramatic regression of esophagitis to grade 0/1 in 22 of 24 patients (91.7% cases), thus underlying the effectiveness of EGCG. On similar lines, a prospective phase II trial confirmed that EGCG can be effective against acute radiation-induced esophagitis as well [121]. Topical administration of EGCG to the radiation field, post-mastectomy and radiotherapy, can resolve radiation dermatitis, as revealed in a phase I study [122].

## 12. Concluding Remarks 

Recent breakthroughs in novel single-cell profiling and spatial transcriptomics have leveraged our understanding to a new level and helped us to find new answers to a critical question of how cancers move through space and time. Importantly, with rapidly increasing sensitivity of detection methods, we also require novel approaches to conceptually analyze single-cell data with observations at the tissue and organ level. 

We have developed a near to complete understanding of VEGF/VEGFR signaling pathways. Studies have shown that relative abundance of the cell surface expression of various VEGFRs and their affiliations for specific VEGF ligands play a fundamental role in the initial set of dimeric constellations. Deeper knowledge of this multifaceted signaling web is key to result-oriented therapeutic targeting. Likewise, EGCG mediated targeting of Wnt/β-catenin has been explored and it needs to be tested comprehensively in different types of cancers. Henceforth xenografted mice bearing β-catenin-overexpressing cancer cells will be helpful in uncovering the true potential of EGCG. Likewise, there is a need to unveil if EGCG inhibited β-catenin activation by functionalization of negative regulators of Wnt signaling. Accordingly, TGF/SMAD signaling regulation by EGCG needs to be addressed more conceptually. Inhibition of SMAD phosphorylation by EGCG is a single dimension of this highly intricate mechanism. Available evidence enlightens involvement of SMURFs and NEDDs in inhibition of TGF/SMAD signaling. Therefore, additional key players of TGF/SMAD signaling also need in-depth research. Regulation of Notch signaling by EGCG seems to be sparsely studied. Therefore, we still have incomplete information about targeting of proteolytically cleaved segment of Notch-intracellular domain (NICD) in regulation of the target gene network. Does EGCG inhibit NICD nuclear accumulation or whether it also interferes with repressor/co-repressor and activator/co-activator machinery needs more answers. On a similar note, SHH/Gli pathway regulation by EGCG requires initial cellular studies. Furthermore, Gli-overexpressing cancers have to be treated with EGCG and combinatorial treatments. 

Despite the absence of clinical trials, the NPs loaded with EGCG might be an efficient and safe strategy for the treatment of several cancers, especially breast and prostate cancer. Thus, clinical trials should be conducted to establish the clinical potential of the NPs loaded with EGCG alone or in addition with the conventional anti-cancer drugs.

## Figures and Tables

**Figure 1 cancers-12-00951-f001:**
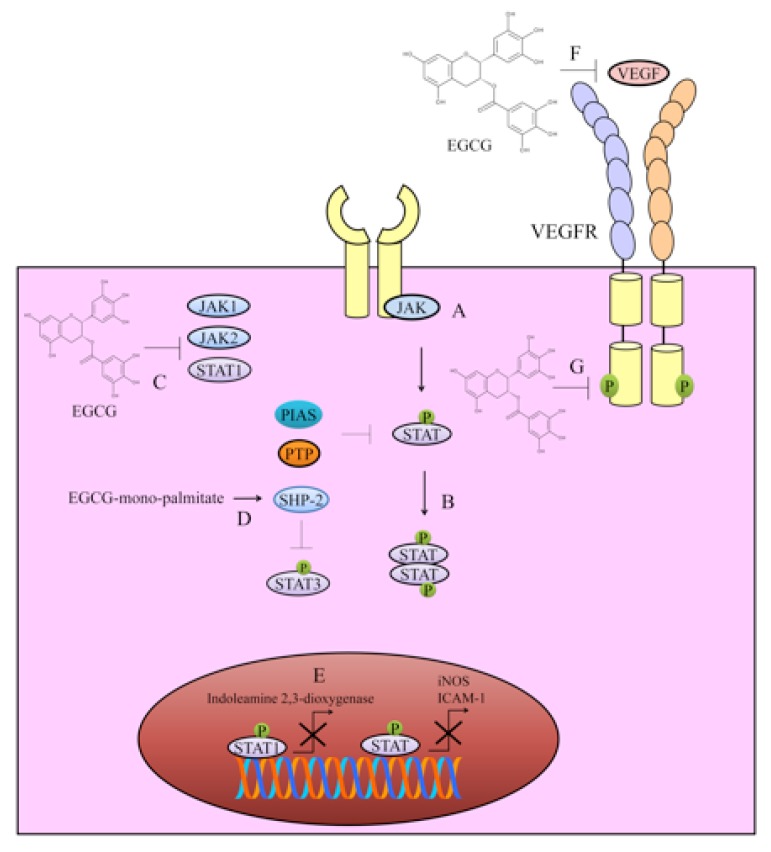
Regulation of the JAK/STAT pathway by epigallocatechin-3-gallate (EGCG). (**A**,**B**) Janus kinase (JAK) mediated phosphorylation of STAT proteins promoted their accumulation in nucleus to regulate expression of a plethora of genes. (**C**–**E**) EGCG showcased remarkable ability to shut down the JAK/STAT pathway by inhibition of Janus kinase 1 (JAK1), Janus kinase 2 (JAK2), signal transducer and activator of transcription 1 (STAT1), signal transducer and activator of transcription 3 (STAT3). EGCG also activated negative regulators of STAT-driven signaling. Activation of Src homology 2 domain-containing tyrosine phosphatase-1 (SHP-2) was effective in inhibition of JAK/STAT signaling. Different oncogenes particularly, inducible nitric oxide synthase (iNOS), intercellular adhesion molecule-1 (ICAM-1), and indoleamine 2,3-dioxygenase have been shown to be under direct control of STAT signaling. (**F**,**G**) Vascular endothelial growth factor vascular endothelial growth factor receptor (VEGF/VEGFR) signaling is also regulated by EGCG. EGCG interacted with VEGF. Additionally, EGCG inhibited phosphorylation of VEGFR.

**Figure 2 cancers-12-00951-f002:**
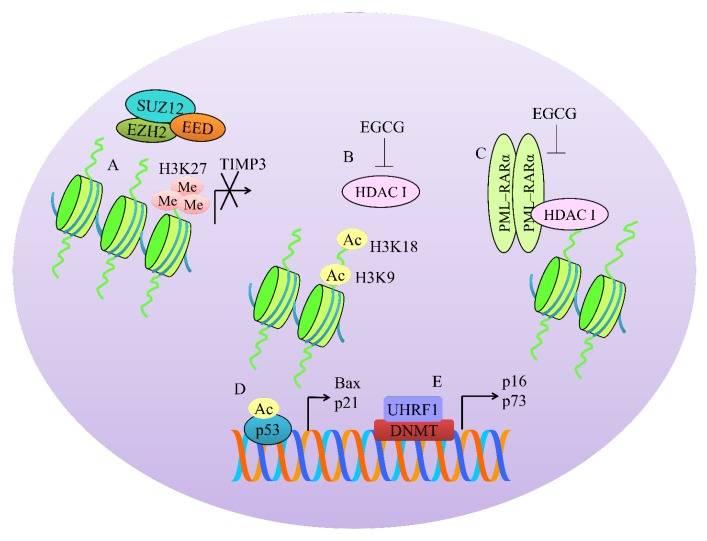
Interconnected and orchestrated interplay among various regulators of epigenetic modifying machinery. (**A**) Protein enhancer of zeste-2 (EZH2), embryonic ectoderm development (EED), and suppressor of zeste 12 (SUZ12) worked synchronously to trimethylate H3K27 and transcriptionally repressed tissue inhibitor of metalloproteinases-3 (TIMP-3). (**B**) Class 1 histone deacetylases (HDACs) were inhibited by EGCG to increase acetylation at H3K9 and H3K18. (**C**) PML–RARα homodimers worked collaboratively with HDAC to regulate expression of target genes. However, EGCG effectively inhibited PML–RARα and HDAC. (**D**) Acetylation of proteins has also been investigated. Acetylated p53 stimulated expression of Bax and p21. (**E**) Ubiquitin-like containing PHD and Ring finger 1 (UHRF1) and DNA methyltransferase (DNMT) also notably downregulated p16 and p73.

**Figure 3 cancers-12-00951-f003:**
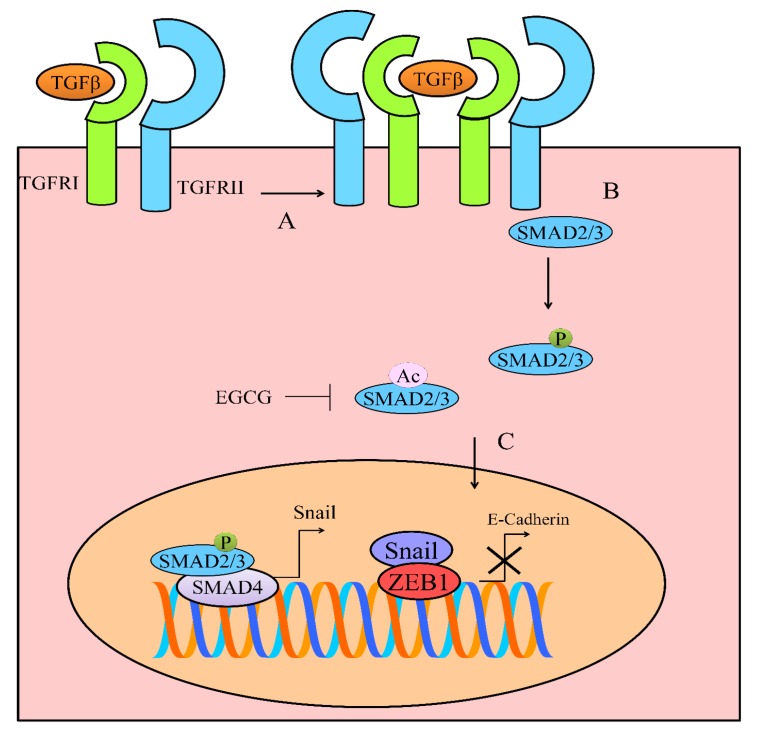
(**A**,**B**) Binding of TGFβ superfamily ligands to a type II receptor induced juxtapositioning of type I receptor. Phosphorylation of SMAD2/3 promoted its accumulation in the nucleus. SMAD2/3 have been shown to stimulate expression of Snail and Slug. Apart from phosphorylation, additional post-translational modifications, particularly acetylation, have also been observed in TGF/SMAD signaling. EGCG inhibited acetylation of SMAD proteins.

**Figure 4 cancers-12-00951-f004:**
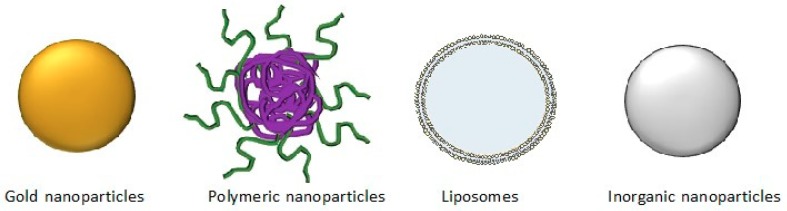
Main types of nanoparticles (NPs) used for the delivery of EGCG.

**Table 1 cancers-12-00951-t001:** Different types of EGCG nanocarriers for cancer management.

Type of Nanoparticles	Route of Administration	Target Organ	Outcome	Ref.
Gold	Oral, intra-tumoral and intra-peritoneal	Bladder	Tumor volume reduction in a bladder xenograft model	[84]
Gold	Intra-tumoral	Skin	Tumor volume reduction in a melanoma cells in a mouse model	[85]
Gold	N/A	Autonomic nervous system	Induction of apoptosis in neuroblastoma cells	[86]
Gold	N/A	Liver	Toxicity in tumor cells and protection of normal mouse hepatocytes	[87]
Polymeric	N/A	Prostate	Toxicity in prostate cancer cell line	[88]
Polymeric	N/A	Colon and rectum	DNA damage levels in samples of lymphocytes from colorectal cancer patients	[89]
Polymeric	N/A	Breast	Toxicity in breast cancer cell line and patient-derived cells	[90]
Polymeric	Intra-tumoral	Prostate	Tumor size reduction in mice model of prostate cancer	[91]
Polymeric	Oral	Prostate	Tumor size reduction in mice model of prostate cancer	[92]
Polymeric	Oral	Skin	Toxicity in human melanoma cells	[93]
Polymeric	N/A	Stomach and intestine	Anti-tumoral activity in gastrointestinal cancer cell line	[94]
Polymeric	N/A	Breast	Inhibition of breast cancer cell line viability	[95]
Lipid-based	Topic and intra-tumoral	Skin	Accumulation of EGCG in the tissues in a mice model of basal cell carcinoma	[96]
Lipid-based	Intra-tumoral	Skin	Apoptosis in a mice model of basal cell carcinoma	[97]
Lipid-based	N/A	Breast	Anti-proliferative and pro-apoptotic effect in a breast cancer cell line	[98]
Lipid-based	N/A	Breast	Cell apoptosis and cell invasion inhibition in a breast cancer cell line	[99]
Sugar-based	N/A	Prostate	Cell viability inhibition in a prostate cancer cell line	[80]
Inorganic	Intra-tumoral	Liver	Tumor growth reduction in a mouse model of liver cancer	[81]
Inorganic	N/A	Prostate	Anti-tumoral activity in prostate cell line	[82]

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
