# Peer review of "EGCG Mediated Targeting of Deregulated Signaling Pathways and Non-Coding RNAs in Different Cancers: Focus on JAK/STAT, Wnt/β-Catenin, TGF/SMAD, NOTCH, SHH/GLI, and TRAIL Mediated Signaling Pathways"

_cancers, 2020, doi:10.3390/cancers12040951_

Round 1

Reviewer 1 Report

In this manuscript Farooqi and colleagues have performed a review of the current literature on the Epigallocatechin-3-gallate (EGCG) focusing on the molecular pathways impaired by this natural compound in different types of cancer.

The manuscript is clearly written and well arranged in the sections and the sub-sections with the consistent layout, however sometimes is easy to get lost because of the density of the information. 

Comments:

In the paragraph 9.2. the Authors should mention and insert this citation: DOI: 10.1038/onc.2010.487

The figures are too basic the Authors should improve them.

Author Response

Referee 1:

Comment 1

“In this manuscript Farooqi and colleagues have performed a review of the current literature on the Epigallocatechin-3-gallate (EGCG) focusing on the molecular pathways impaired by this natural compound in different types of cancer. The manuscript is clearly written and well arranged in the sections and the sub-sections with the consistent layout, however sometimes is easy to get lost because of the density of the information. Comments: In the paragraph 9.2. the Authors should mention and insert this citation: DOI: 10.1038/onc.2010.487”.

We acknowledge the referee comment and we are in full agreement. Thus, reference has been added.

Comment 2

“The figures are too basic the Authors should improve them”.

We completely agree with the referee and, therefore we have re-designed figure 1 for a better outlook and representation of the data.

Reviewer 2 Report

Farooq et al have very well explained the regulation of JAK/STAT, Wnt/β-Catenin, TGF/SMAD, SHH/GLI, NOTCH pathways as well as modulation of non-coding RNAs in different cancers by EGCG.

Considering the significant potential of EGCG, I would suggest to add a paragraph about the clinical trials.

Author Response

Referee 2:

Comment 1

“Farooq et al have very well explained the regulation of JAK/STAT, Wnt/β-Catenin, TGF/SMAD, SHH/GLI, NOTCH pathways as well as modulation of non-coding RNAs in different cancers by EGCG. Considering the significant potential of EGCG, I would suggest to add a paragraph about the clinical trials”.

We completely agree with the referee and, therefore, we have inserted new information in the manuscript. Additionally, we have improved the section of concluding remarks for a better take-home message for our readers.

New information inserted (Line 529-563):

“Clinical trials evaluating EGCG

Given the promising reportings from pre-clinical studies, EGCG has been tested in various clinical studies. Postmenopausal women are at high risk of developing breast cancer and, therefore, EGCG safety clinical trials have been conducted targeting this population. EGCG can afford benefit in terms of regulating LDL-cholesterol as well as glucose and insulin, as reported by a double-blind, randomized, placebo-controlled intervention study in healthy postmenopausal women [112]. A subsequent ancillary study of a double-blind, randomized, placebo-controlled, parallel-arm trial further confirmed the benefit of EGCG but reported the total cholesterol levels reductions only in women with elevated baseline levels [113]. In postmenopausal women, a daily dose of 843 mg EGCG has been reported to be generally well-tolerated with only a small fraction (6.7%) of women reporting adverse events [114]. This dose of 843 mg EGCG, when administered for a year, can reduce mammographic density in relatively younger women (50-55 years) but not in postmenopausal women, as suggested by phase II trial [115]. Not only in breast cancer patients or women at high breast cancer risk, EGCG is well-tolerated by chronic lymphocytic leukemia (CLL) patients as well [116]. Further, EGCG, at a daily dose as low as 44.9 mg for 4 weeks prior to surgery, has been reported to result in increased bioavailability,including accumulations in breast tumour tissue, in early stage breast cancer patients [108].

A randomized trial reported no reduction in likelihood of prostate cancer in men with high-grade prostatic intraepithelial neoplasia, compared to placebo, after a year on 400 mg EGCG dose per day [117]. It is possible that this might be related to the dose tested in this study as a previous study which tested the effects of 800 mg EGCG administered to twenty-six patients with positive prostate biopsies reported significant reductions in PSA, HGF and VEGF, with no associated liver toxicity [118]. Similarly, a phase II pharmacodynamic prevention trial in bladder cancer patients indicated a possible reduction in PCNA and clusterin levels upon 2-4 weeks administration of EGCG prior to transurethral resection of bladder tumour or cystectomy [119].

EGCG has been tested in cancer clinical trials not just for the direct anticancer effects, but also for possible effects on co-morbidities. In lung cancer patients with an unresectable stage III disease, a phase I study was conducted to evaluate the efficacy of EGCG against chemotherapy related esophagitis [120]. Patients, divided in six cohorts receiving six different doses of EGCG, were administered EGCG once grade 2 esophagitis occurred. The study reported dramatic regression of esophagitis to grade 0/1 in 22 of 24 patients (91.7% cases), thus underlying the effectiveness of EGCG. On similar lines, a prospective phase II trial confirmed that EGCG can be effective against acute radiation-induced esophagitis as well [121]. Topical administration of EGCG to the radiation field, post-mastectomy and radiotherapy, can resolve radiation dermatitis, as revealed in a phase I study [122]”.

Reviewer 3 Report

An interesting and informative review is presented for review, giving fairly detailed information about the mechanisms of antitumor activity of EGCG. However, the review structure can be adjusted. in particular, in my opinion, information is easier to perceive if it would be structured into sections (for example): activation of apoptosis, effect on angiogenesis and invasion, and effect on intracellular transduction of mitogenic signals. But this is at the discretion of the authors.

Author Response

We acknowledge the referee comment.